# Development of HIV Drug Resistance in a Cohort of Adults on First-Line Antiretroviral Therapy in Tanzania during the Stavudine Era

Raphael Z. Sangeda [1,2,*], Perpétua Gómes [3,4], Soo-Yon Rhee [5], Fausta Mosha [2,6], Ricardo J. Camacho [2], Eric Van Wijngaerden [7], Eligius F. Lyamuya [8] and Anne-Mieke Vandamme [2,9]

1   Department of Pharmaceutical Microbiology, Muhimbili University of Health and Allied Sciences, Dar es Salaam P.O. Box 65013, Tanzania

2   Department of Microbiology, Immunology and Transplantation, Rega Institute for Medical Research, KU Leuven-University of Leuven, 3000 Leuven, Belgium; fausta_mosha@yahoo.com (F.M.); Ricardo.camacho@mybiodata.eu (R.J.C.); annemie.vandamme@uzleuven.be (A.-M.V.)

3   Centro Investigação Interdisciplinar Egas Moniz (CiiEM), 2829-511 Caparica, Portugal; pcrsilva@chlo.min-saude.pt

4   Molecular Biology Laboratory, Centro Hospitalar de Lisboa Ocidental, 1349-019 Lisbon, Portugal

5   Division of Infectious Diseases, Stanford University, Stanford, CA 94305, USA; syrhee@stanford.edu

6   Ministry of Health, Community Development, Gender, Elderly and Children, Dar es Salaam P.O. Box 3448, Tanzania

7   Department of Microbiology, Immunology and Transplantation and University Hospitals, KU Leuven, 3000 Leuven, Belgium; eric.vanwijngaerden@uzleuven.be

8   Department of Microbiology and Immunology, Muhimbili University of Health and Allied Sciences, Dar es Salaam P.O. Box 65001, Tanzania; eligius_lyamuya@yahoo.com

9   Center for Global Health and Tropical Medicine, Unidade de Microbiologia, Instituto de Higiene e Medicina Tropical, Universidade Nova de Lisboa, 1349-008 Lisbon, Portugal

*   Correspondence: sangeda@gmail.com

**Abstract:** As more HIV patients start combination antiretroviral therapy (cART), the emergence of HIV drug resistance (HIVDR) is inevitable. This will have consequences for the transmission of HIVDR, the success of ART, and the nature and trend of the epidemic. We recruited a cohort of 223 patients starting or continuing their first-line cART in Tanzania towards the end of the stavudine era in 2010. Patients were then followed for one year. Of those with a viral load test at baseline and follow-up time, 34% had a detectable viral load at the one-year endpoint. For 41 patients, protease and reverse transcriptase genotyping were successful. Eighteen samples were from cART-naïve patients, and 23 samples were taken under therapy either at baseline for cART-experienced patients or from follow-up samples for both cART–naïve and cART–experienced patients. The isolates were subtype A, followed by C and D in 41.5%, 22%, and 12.2% of the patients, respectively. No transmitted HIVDR was detected, as scored using the surveillance drug resistance mutations (DRMs) list. However, in 3 of the 18 samples from cART-naïve patients, the clinical Rega interpretation algorithm scored 44D or 138A as non-nucleoside reverse transcriptase inhibitor (NNRTI) resistance-associated polymorphisms. The most observed nucleoside reverse transcriptase inhibitor (NRTI) mutation was 184V. The mutation was found in 16 patients, causing resistance to lamivudine and emtricitabine. Nineteen patients had NNRTI resistance mutations, the most common of which was 103N, observed in eight patients. These high levels of resistance call for regular drug resistance surveillance in Tanzania to inform the control of the emergence and transmission of HIVDR.

**Keywords:** HIV-1; drug resistance; Dar es Salaam; Tanzania; NNRTI; NRTI

## 1. Introduction

The recent scale-up of combination antiretroviral therapy (cART) in resource-limited settings (RLS) has significantly reduced morbidity and mortality among HIV and AIDS

patients. The success of these programs stems from the population-based approach to provide affordable and simplified standard first- and second-line regimens recommended by the World Health Organization (WHO). Among the few ARVs that are available in such settings, a combination of two nucleoside reverse transcriptase inhibitors (NRTIs) and one non-nucleoside reverse transcriptase inhibitor (NNRTI) is used as the first-line regimen [1]. In a recent update, the WHO recommendations included a more potent dolutegravir (DTG) belonging to the class of integrase strand transfer inhibitors (INSTIs), along with two NRTIS backbones [2]. The standard second-line regimen before DTG recommendation was lopinavir or atazanavir boosted with ritonavir as the protease inhibitor (PI), recommended with 2 NRTIs. The main concern with these costly treatment programs is that they can compromise the utility of the first-line regimen by i) the low genetic barrier to resistance of NNRTIs [3], ii) long-term side effects such as toxicity, lipodystrophy and peripheral neuropathy that are associated with the use of stavudine [4], one of the main NRTI components of first-line therapy in many RLS, which increases the chances of non-adherence [5], and iii) failure of first-line regimens due to lack of potency of ARV combinations, insufficient drug adherence and transmission of drug-resistant strains [6]. Although many countries have scaled up the use of tenofovir, thymidine analogs, such as stavudine or zidovudine, are still used in Sub-Saharan Africa [7,8].

In developed countries, the standard of care is to change treatment when the viral load becomes detectable and to guide the next line of therapy by assessing the susceptibility of patient isolates using genotypic assays to select ARV drugs, which can bring a successful treatment response. Several publicly available algorithms [9–12] are used to interpret the mutations. Prospective controlled studies have shown that patients whose physicians have access to HIV drug-resistance (HIVDR) data, particularly genotypic resistance data, respond better to therapy than control patients of physicians without such access [10,13]. These kinds of data have led several experts in North America and Europe to recommend HIVDR testing to manage HIV-1-infected patients [10,14,15]. In RLS, individuals are currently monitored using clinical and immunological criteria only because of the high cost of viral load assays and HIVDR genotyping. The surveillance of HIVDR is monitored through population-based surveys of early-warning indicators (EWI) predefined by the WHO. Factors monitored as EWI include antiretroviral therapy (ART) prescribing practices; patients lost to follow-up after initiation of ART; patients on appropriate first-line treatment at 12 months; on-time patient appointment keeping and ARV drug pick-ups; and ARV drug-supply continuity. Optionally, other adherence measurements and HIV viral load suppression at 12 months may be collected [16].

In Tanzania, patients return to care and treatment centers every three to six months for ARV refills and medical evaluation based on clinical symptoms and immunological progress. A few genotypic resistance tests have only recently been conducted [17–20], and the extent of resistance development during cART in Tanzania is mostly unknown. The objective of this study was to document the development of HIVDR during first-line therapy in Tanzania. We determined the HIV-1 protease and reverse transcriptase genotypic diversity and drug resistance mutations (DRMs) at study baseline and one-year follow-up, selected from our cohort and for whom we had viral load measurements at study baseline or one year of follow-up.

## 2. Materials and Methods

### 2.1. The Cohort Description

Patients were recruited into the study between May and July 2010. Samples were collected during a prospective cohort study involving first-line ARV users at Amana District Hospital Care and Treatment Center (CTC) in the Ilala Municipality in Dar es Salaam, Tanzania, as previously described [21,22]. A total of 254 randomly chosen patients was consulted for inclusion in the cohort (as described previously and in Figure 1). Selection criteria were either starting cART or being on a first-line ART for more than four months. Exclusion criteria were being below 18 years, pregnant, having opportunistic infections

or malignancy. A total of 31 patients was excluded due to various reasons. A total of 223 patients was finally included, 26 of whom were cART-naïve and 197 of whom were cART-experienced for more than four months at enrolment. The follow-up samples were collected between May and July 2011.

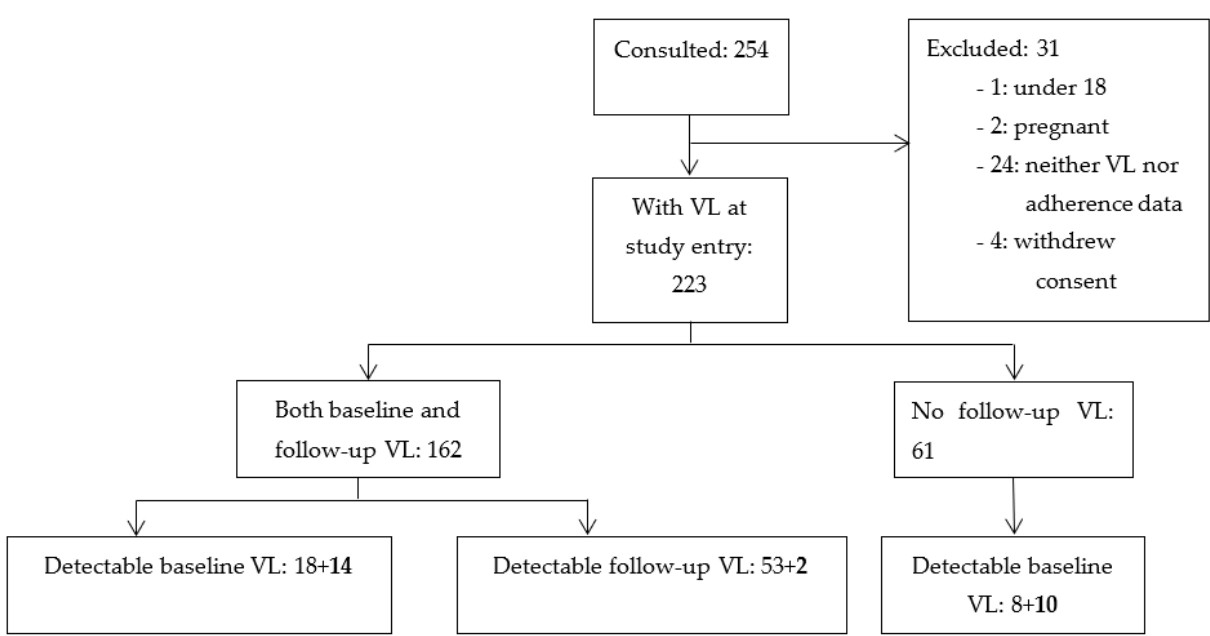

**Figure 1.** Patient sampling flowchart. Starts from patients invited to participate in the study cohort. A total of 105 patients with detectable viral load (VL) are shown in the boxes at the bottom. In bold is the number of patients who started therapy at study entry, and the others were already cART-experienced patients.

*2.2. Data Collection Procedures*

2.2.1. Treatment History and Clinical Data

The patients' treatment and clinical data were collected using a patient's history from manual and electronic medical records.

2.2.2. Drawing of Blood Samples

For viral load testing and genotyping, 10 ml of blood was collected in EDTA tubes from each patient at the study baseline and one year later. The samples collected at study entry were from both cART-naïve and cART-experienced patients, while only cART-experienced patients were sampled at one-year follow-up. Plasma samples were separated from cells by centrifugation and frozen at $-70\,^{\circ}$C within 24 h. These samples were kept at the Laboratory of Microbiology and Immunology, Muhimbili University of Health and Allied Sciences (MUHAS), Dar es Salaam, Tanzania.

2.2.3. Viral Load Measurement

Viral load testing was done using the Roche Taqman 2.0 system, the only assay available in the Microbiology and Immunology (MUHAS) laboratory at the time of the study, which has a detection limit of $\leq$400 copies/mL. The baseline data were obtained at enrollment into the study, and follow-up data were recorded one year later after enrollment into the study. Virological failure was defined as having a viral load above the detection limit of 400 copies/mL.

2.2.4. Genotyping of Patients' HIV Isolates

In November 2011, the samples were shipped to the Molecular Biology Laboratory, Centro Hospitalar de Lisboa, Portugal for genotyping. In total, 105 samples (Figure 1) had a detectable viral load at any one time. A total of 95 samples from 82 patients were sent

for genotyping after an error in packing ten samples during shipping. HIV-1 genotyping was performed with the ViroSeq HIV-1 Genotyping System (Abbot Diagnostics) or an in-house system [23]. Protease (PR) and reverse transcriptase (RT) nucleotide sequences were analyzed with an ABI PRISM 3100 automated sequencer. The sequences were deposited into GenBank with accession numbers MN816754-MN816797.

### 2.3. Data Storage and Analysis

RegaDB software [24] was used to store patient data, along with the viral sequences. Data included patient treatment history indicating the duration of therapy and the actual drugs taken by each patient. Built-in tools were used to identify the HIV-1 subtypes and circulating recombinant forms (CRFs), amino acid mutation lists, and drug resistance profiles. Subtypes were identified based on the Rega subtyping tool 3.0 [25]. Transmitted DRMs were assessed using the WHO updated surveillance DRMs [26] and Stanford's calibrated population resistance tool (CPR). Genotypic susceptibility scores (GSS) were based on Rega HIVDR interpretation algorithm version 8.0.1 [9], implemented in the RegaDB software. Each drug received a GSS score of 1 if susceptible, except boosted PI, which scored 1.5; 0.5 if intermediate resistance for an NRTI; 0.25 for an NNRTI; and 0.75 for a boosted PI. An isolate scored 0 if resistant to the drug. Total GSS scores for the regimen used at the time of sampling and the available second-line regimens were calculated as the individual drugs' cumulated score.

Data stored in RegaDB was exported into an R statistical software package for further analysis, including descriptive statistics and mutation tables. Descriptive analyses including median, interquartile range (IQR) for numerical variables, frequencies, and proportions for categorical variables were performed and tested for association using Fisher or Chi-square tests. For continuous variables, the Wilcoxon signed-rank test or Mann-Whitney's test for continuous values was used to test associations. HIVDR was defined as the presence of one DRM out of the following list [27,28]: RT 41L, 44D, 62V, 65R, 67N, 70R, 74I/V, 75I, 77L, 100I, 101P, 103N/S, 106A/M, 108I, 115F, 116Y, 118I, 138A/G/K/Q/R, 151M, 179L, 181C/V, 184I/V, 188C/H/L, 190A/S, 210W, 215F/Y, 219E/Q, 221Y, 225H, 227C and 230I/L for RT; protease10I/F/V, 11I, 16E, 20I/M/R/T/V, 24I, 30N, 33I/F/V, 34Q, 36I/L/V, 43T, 46I/L, 47A/V, 48V, 50V, 53L/Y, 54L/V, 54M/T/A, 60E, 62V, 63P, 64L/M/V, 69K/R, 71I/L/T/V, 73A/C/S/T, 74P, 76V, 77I, 82A/F/S/T, 84V, 85V, 88D/S, 89I/M/V, 90M and 93L/M.

The cut-off level of significance for all analyses was $p < 0.05$. All statistical analyses were performed using the R-statistical package version 2.15.1 [29].

## 3. Results

### 3.1. Description of Cohort Regimens

For those patients who were already on their first-line treatment at the start of the study, the distribution of various therapy regimens was as follows. A fixed combination of a twice-a-day dose of Triomune-30 and a coformulation of stavudine (d4T), lamivudine (3TC), and nevirapine (NVP) was the commonly dispensed therapy to 101 (45.9%) of all patients. One patient received d4T + 3TC + efavirenz (EFV), and 97 patients (44.1%) were on Combivir- (zidovudine (AZT) + 3TC) based therapy in combination with EFV, NVP, or abacavir (ABC) in 54, 42, and one patient(s), respectively. During the one-year follow-up, 13 patients had switched therapy for reasons of toxicities to the ARVs (Supplementary Table S1).

### 3.2. The Success Rate of Genotyping

Of the 105 samples with detectable viral load, 95 samples from 82 patients were available for genotyping, 47 from the study baseline and 48 from follow-up samples. Genotyping was successful in 44 of the 95 samples (46.3%) obtained from 41 of the 82 patients. Of the successful samples, 18 were baseline samples from cART-naïve patients (Table 1), and 26 were from 23 cART-experienced patients (Supplementary Table S1) for more than four

months (11 baseline samples, 15 follow-up samples, with only two patients both baseline and follow-up samples: patient numbers 27 and 35). Two of the 18 cART-naïve patients were virologically failing at one-year follow-up periods (Supplementary Table S1).

### 3.3. Subtype Diversity

The HIV subtype distribution of the isolates is shown in Figure 2. Subtype A was the dominant subtype in 41.5% of the patients, followed by C and D at 22% and 12.2%, respectively.

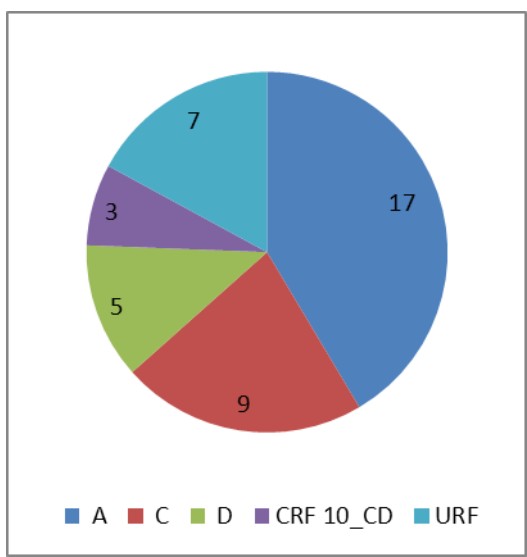

**Figure 2.** Subtype distribution of isolates from 41 patients. The numbers represent the isolates per subtype. CRF = circulating recombinant form. URF = unique recombinant form, consisted of CRF 10 CD_D (3), A1_C (1), C_D (1) and D_A1 (2) recombinants.

### 3.4. Virological Response Data

At one year of follow-up, longitudinal viral load measurements were available for 162 patients (Figure 1). The virological response for this set of patients with viral load measurement at both study baseline and follow-up of the study is summarized in Table 2. Briefly, of the 162 patients, 14 were cART-naïve at recruitment, and all had a detectable viral load. Of the 148 with therapy experience at recruitment, 18 (12.2%) had a detectable viral load. At one year follow-up, 55 (34%) of the 162 patients had a detectable viral load. A total of 15 patients had a detectable viral load at both time points, only two of whom were cART-naïve.

Taking only the 210 patients who were on therapy for at least six months, a total of 354 viral load measurements were available. These patients had been on treatment for a median (IQR) of 32 (22–44) months. A total of 80 (22.6%) patients had a detectable viral load (see Figure 3). The median (IQR) duration of therapy in the various time windows was 10 (8.25–11), 18 (15–22), 31 (28–34), and 48 (41–56) months for the one, two, three and more than three years groups, respectively. The proportion of patients with detectable viral load was: 28.57% in year 1, 13.86% in year 2, 30.39% in year 3, and 22.63% of those on therapy for more than three years. There was a significant positive correlation in the proportion of patients with detectable viral load with increasing exposure to therapy ($p$-value = 0.03) (Figure 3).

**Table 1.** Characteristics of patients and the mutations found in the RT and PR region from HIV isolates of the cART-naïve patients.

| Patient # | Sequence ID | Age at Study Entry | Gender | Subtype † | Pharmacy Refill Adherence (%) | Therapy * (in Addition to AZT+3TC) | Log VL at Study Baseline | Log VL at one Year | NNRTI Resistance-Related Polymorphisms | PI Resistance-Related Polymorphisms |
|---|---|---|---|---|---|---|---|---|---|---|
| 1 | N001 | 29 | F | CRF 10_CD | NA | EFV | 4.74 | NA | None | 36I, 63P |
| 2 | N002 | 32 | F | A | 100 | EFV | 5.94 | ≤2.60 | None | 10I, 36I, 69K, 89M |
| 3 | N003 | 26 | F | A | 100 | EFV | 5.81 | NA | 138A | 36I, 62V, 69K, 89M |
| 4 | N004 | 35 | M | A | 77.1 | EFV | 5.06 | NA | 138A | 20I, 36I, 64M, 69K, 89M |
| 5 | N006 | 35 | F | A | 95.84 | NVP | 5.12 | ≤2.60 | 44D | 36I, 69K, 89M |
| 6 | N007 | 35 | M | URF | 100 | NVP | 3.31 | NA | None | 36I, 69K, 89M |
| 7 | N008 | 30 | F | A | 93.29 | NVP | 4.79 | 4.72 | None | 10I, 16E, 36I, 69K, 77I, 89M |
| 8 | N010 | 34 | F | A | 81.02 | EFV | 4.32 | ≤2.60 | None | |
| 9 | N011 | 42 | M | D | 91.47 | EFV | 6.00 | ≤2.60 | None | 10V, 63P, 64V |
| 10 | N012 | 38 | F | URF | 93.38 | EFV | 5.74 | ≤2.60 | None | 64V |
| 11 | N013 | 54 | M | D | 100 | EFV | 4.23 | NA | None | 64M, 77I |
| 12 | N015 | 55 | M | A | 97.39 | NVP | 5.47 | ≤2.60 | None | 11I, 36I, 63P, 69K, 89M/I |
| 13 | N016 | 52 | M | C | 92.57 | EFV | 5.65 | ≤2.60 | None | 36I, 89M, 93L |
| 14 | N017 | 32 | M | C | 100 | EFV | 4.93 | ≤2.60 | None | 36I, 69K, 89M, 93L |
| 15 | N018 | 33 | F | D | 100 | EFV | 5.05 | ≤2.60 | None | 64V, 77I |
| 16 | N019 | 32 | M | C | 48.19 | EFV | 5.56 | NA | None | 16E, 36I, 69K, 89M, 93L |
| 17 | N020 | 56 | F | A | 70.45 | EFV | 5.58 | NA | None | 20R, 36I, 64L, 69K, 89M |
| 18 | N022 | 31 | F | A | 84.87 | EFV | 5.16 | ≤2.60 | None | 10I, 11I, 36I, 63P, 69K, 89M |

* None of these patients changed regimen for the entire follow-up period. † URF: unique recombinant forms, see Figure 2. NNRTI = non-nucleoside reverse transcriptase inhibitor; PI = protease inhibitor; NA = not available; 3TC = lamivudine, AZT = zidovudine, EFV = efavirenz, NVP = nevirapine. No nucleoside reverse transcriptase inhibitor resistance mutation was found; genotypic susceptibility score for all patients was 3.

**Table 2.** Viral load among study patients, listed according to therapy experience for 162 patients with both study baseline and follow-up viral load pair.

| | | | Detectable Viral Load | | | |
|---|---|---|---|---|---|---|
| | | | At Study Entry | | At One Year Follow-Up | |
| Time of Therapy Initiation | Number of Patients | Duration of Therapy at Study Entry Median Months (IQR) | N (%) | Log VL Median (IQR) | N (%) | Log VL Median (IQR) |
| At study entry | 14 | 0 | 14 (100%) | 5.1 (4.9–5.6) | 2 (14.3%) | 4.4 (4.1–4.6) |
| Before study entry | 148 | 25 (18–36) | 18 (12.2%) | 4.4 (3.5–4.7) | 53 (35.8%) | 3.9 (2.9–4.8) |

VL = Viral load; IQR = Interquartile range.

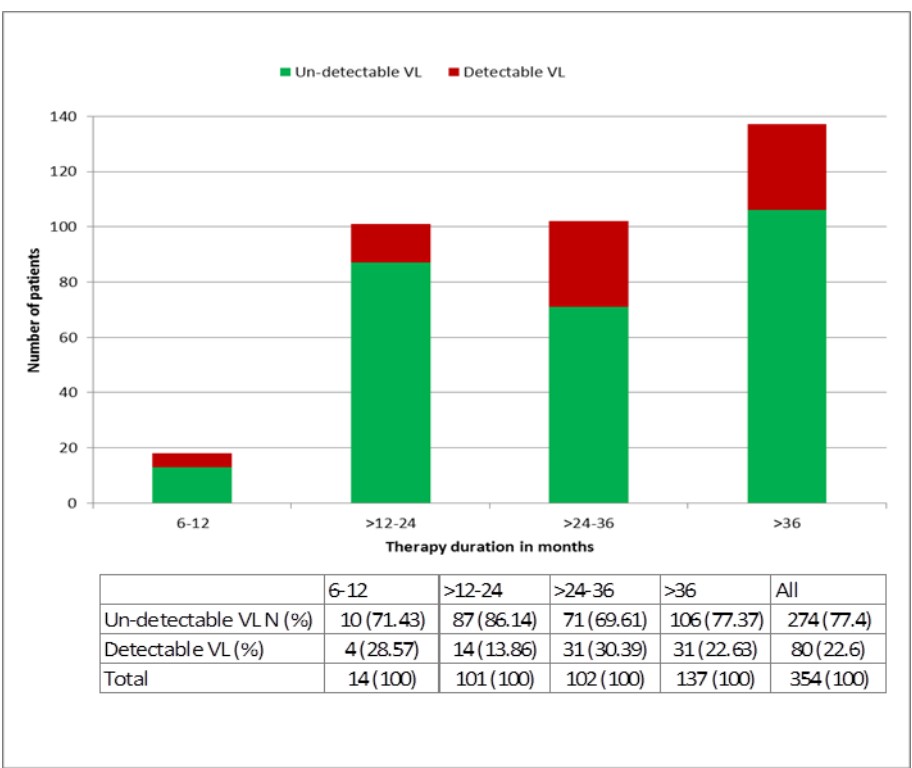

**Figure 3.** The proportion of patients with successful treatment (undetectable viral load) at different time intervals after the start of first-line treatment. (Undetectable viral load on therapy can be considered successful treatment only for patients ≥ 6 months on therapy, EACS guidelines [30]).

### 3.5. HIV Drug Resistance

3.5.1. Pre-Treatment Drug Resistance

For the analyzed patients, the genotypic resistance profile and therapy changes during follow-up are shown in Table 1 and Supplementary Table S1. No transmitted HIVDR was detected among the 18 available genotypes in patients that were starting therapy at recruitment. However, 44D or 138A RT resistance-associated polymorphisms scored by the Rega algorithm were detected in three patients. No genotype at follow-up was available for patients that initiated therapy at recruitment; they all had either an undetectable viral load or a low viral load (Supplementary Table S1). Considering baseline and follow-up samples together, at least one DRM (excluding PI polymorphisms) was observed in 19 (82.6%) of the 23 therapy-experienced patients with genotyping results. NNRTI and NRTI mutations were found in the baseline sample of 19 and 16 of these patients, respectively. Dual NNRTI and NRTI mutations were observed in 16 patients.

### 3.5.2. Acquired Drug Resistance

Considering all samples together, the most frequently observed RT mutation was 184V (Table 3 and Supplementary Table S1), followed by 103N. No major PI mutation was found in any of the samples, but all except one patient harbored some minor PI mutations, which is expected, considering the subtypes. For the two patients with baseline and follow-up genotypes, resistance evolution was observed. In one, the mutations 184V and 190A occurred first, followed by the accumulation of TAMs (41L, 67N, 70R, 75I, and 215F). In the second patient, mutations 67N, 70R, 181C, and 184V were observed first, followed by 215F and 219E. All observed resistance mutations were related to the therapy received by the respective patients. The polymorphisms 44D and 138A were not observed in cART-experienced patients.

**Table 3.** Prevalence of resistance mutations or natural polymorphisms scored as related to resistance in reverse transcriptase and protease regions among cART-experience patients. Only the last sample was counted if more than one sample was available. NRTI = nucleoside reverse transcriptase inhibitor; NNRTI = non-nucleoside reverse transcriptase inhibitor; PI = Protease Inhibitor.

| NRTI Mutations | N | % | NNRTI Mutations | N | % | PI Polymorphisms | N | % |
|---|---|---|---|---|---|---|---|---|
| 184V | 17 | 70.8 | 103N | 8 | 33.3 | 36I | 20 | 83.3 |
| 67N | 7 | 29.2 | 181C | 7 | 29.2 | 69K | 19 | 79.2 |
| 70R | 6 | 25.0 | 190A | 5 | 20.8 | 89M | 15 | 62.5 |
| 215F | 5 | 20.8 | 108I | 3 | 12.5 | 20R | 10 | 41.7 |
| 219E | 3 | 12.5 | 138Q | 2 | 8.3 | 16E | 8 | 33.3 |
| 219Q | 2 | 8.3 | 221Y | 2 | 8.3 | 93L | 7 | 29.2 |
| 215Y | 2 | 8.3 | 118I | 1 | 4.2 | 63P | 5 | 20.8 |
| 41L | 2 | 8.3 | 138A | 1 | 4.2 | 64V | 4 | 16.7 |
| 75I | 2 | 8.3 | 179L | 1 | 4.2 | 10V | 2 | 8.3 |
| 151M | 1 | 4.2 | 181I | 1 | 4.2 | 10I | 1 | 4.2 |
| 116Y | 1 | 4.2 | 181V | 1 | 4.2 | 36L | 1 | 4.2 |
| 210W | 1 | 4.2 | 225H | 1 | 4.2 | 62V | 1 | 4.2 |
| | | | | | | 64L | 1 | 4.2 |
| | | | | | | 89I | 1 | 4.2 |

The Rega algorithm scored a median genotypic susceptibility score (GSS) of 3 for isolates from cART-naïve patients. For cART-experienced patients, the median GSS (IQR) was 1.0 (0.5–2.0) (Table 4). Most patients had NNRTI resistance, both against NVP and EFV (Figure 4).

A large proportion of patients had resistance to NRTIs: 69.6%, 69.6%, 21.7%, 21.7%, and 8.7% of the treated patients with genotyping results conveyed resistance to the drugs lamivudine (3TC), emtricitabine (FTC), zidovudine (AZT), stavudine (D4T), and abacavir (ABC), respectively (Table 4). The median (IQR) calculated GSS for standard second-line ABC + FTC + boosted lopinavir (LPV/r) and tenofovir disoproxil fumarate (TDF) + FTC + LPV/r therapies was 2.5 (2.0–3.5) and 2.5 (2.5–3.5), respectively (Table 4).

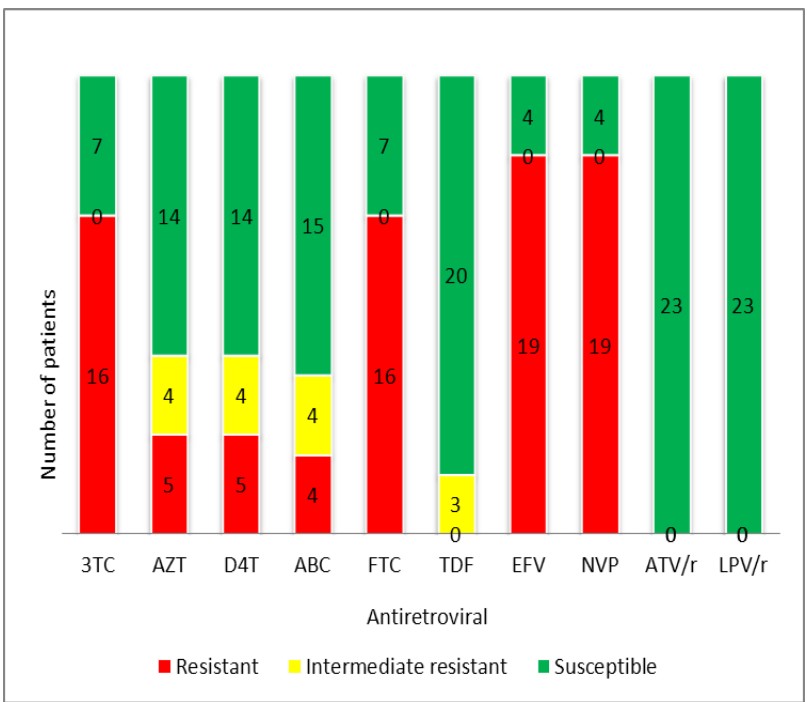

**Figure 4.** Proportion of drug resistance to antiretrovirals among successfully genotyped patients failing their first-line regimen in Tanzania. Genotypic interpretation according to Rega algorithms (v 8.0.1). 3TC = lamivudine, ABC = abacavir, AZT = zidovudine, D4T = stavudine, FTC = emtricitabine, TDF = tenofovir disoproxil fumarate, EFV = efavirenz, NVP = nevirapine, ATV/r = boosted atazanavir, LPV/r = boosted lopinavir.

**Table 4.** Genotypic susceptibility score. Genotypic susceptibility of the current regimen and expected second-line regimen in genotypes from cART-experienced patients.

| | | | | | GSS of Regimen | GSS of Potential Second-Line | |
|---|---|---|---|---|---|---|---|
| Patient # | Sequence ID | Resistance to NRTIs | Intermediate Resistance to NRTIs | Resistance to NNRTIs | At Sampling | ABC + FTC + LPV/r | TDF + FTC + LPV/r |
| 19 | F001 | None | none | none | 3 | 3.5 | 3.5 |
| 20 | W0141 | None | none | none | 3 | 3.5 | 3.5 |
| 21 | W0067 | None | none | none | 3 | 3.5 | 3.5 |
| 22 | F110 | None | AZT, D4T | none | 3 | 3.5 | 3.5 |
| 23 | F189 | None | none | EFV, NVP | 2 | 3.5 | 3.5 |
| 24 | F134 | None | none | EFV, NVP | 2 | 3.5 | 3.5 |
| 25 | F141 | None | none | EFV, NVP | 2 | 3.5 | 3.5 |
| 26 | F144 | 3TC, FTC | none | EFV, NVP | 1 | 2.5 | 2.5 |
| 27 | F176 | 3TC, FTC | none | EFV, NVP | 1 | 2.5 | 2.5 |
| 27 | W0037 | 3TC, AZT, D4T, FTC | ABC | EFV, NVP | 0 | 2 | 2.5 |
| 27 | W0116 | 3TC, ABC, AZT, D4T, FTC | TDF | EFV, NVP | 0 | 1.5 | 2 |
| 28 | W0065 | 3TC, FTC | none | EFV, NVP | 1 | 2.5 | 2.5 |
| 29 | W0021 | 3TC, FTC | none | EFV, NVP | 1 | 2.5 | 2.5 |
| 30 | F064 | 3TC, FTC | none | EFV, NVP | 1 | 2.5 | 2.5 |
| 31 | W0079 | 3TC, FTC | none | EFV, NVP | 1 | 2.5 | 2.5 |
| 32 | W0127 | 3TC, FTC | none | EFV, NVP | 1 | 2.5 | 2.5 |
| 33 | W0019 | 3TC, FTC | none | EFV, NVP | 1 | 2.5 | 2.5 |

**Table 4.** *Cont.*

| Patient # | Sequence ID | Resistance to NRTIs | Intermediate Resistance to NRTIs | Resistance to NNRTIs | GSS of Regimen | GSS of Potential Second-Line | |
|---|---|---|---|---|---|---|---|
| | | | | | At Sampling | ABC + FTC + LPV/r | TDF + FTC + LPV/r |
| 34 | W0066 | 3TC, FTC | none | EFV, NVP | 1 | 2.5 | 2.5 |
| 35 | F003 | 3TC, FTC | ABC, AZT, D4T | EFV, NVP | 0.5 | 2 | 2.5 |
| 35 | W0108 | 3TC, ABC, AZT, D4T, FTC | TDF | EFV, NVP | 0 | 1.5 | 2 |
| 36 | W0158 | 3TC, ABC, AZT, D4T, FTC | TDF | EFV, NVP | 0 | 1.5 | 2 |
| 37 | F068 | 3TC, FTC | ABC, AZT, D4T | EFV, NVP | 0.5 | 2 | 2.5 |
| 38 | W0120 | 3TC, FTC | ABC, AZT, D4T | EFV, NVP | 0.5 | 2 | 2.5 |
| 39 | W0054 | 3TC, ABC, AZT, D4T, FTC | none | EFV, NVP | 0 | 1.5 | 2.5 |
| 40 | W0062 | 3TC, D4T, FTC | ABC, AZT | EFV, NVP | 1 | 2 | 2.5 |
| 41 | F183 | 3TC, AZT, FTC | ABC, D4T | EFV, NVP | 0.5 | 2 | 2.5 |

NA = Not applicable/available; NRTI = nucleoside reverse transcriptase inhibitor; NNRTI = non-nucleoside reverse transcriptase inhibitor. 3TC = lamivudine, ABC = abacavir, AZT = zidovudine, D4T = stavudine, FTC = emtricitabine, TDF = tenofovir disoproxil fumarate, EFV = efavirenz, NVP = nevirapine, ATV/r = boosted atazanavir, LPV/r = boosted lopinavir. Samples F176, W0037, and W0116 from patient number 27; the first was taken at study entry and the last two after one-year follow-up at the one-month interval.

## 4. Discussion

In this prospective cohort study in Tanzania, we followed patients on first-line treatment and reported primary and acquired drug resistance. Although the design of the study was prospective, the failure to obtain a genotype for 50% of our viral-load positive samples and successes heavily biased towards samples with higher viral load means that we can merely report on the resistance evolution found.

The most prevalent genotypes among the isolates from Tanzanian patients were subtype A, followed by C and D. This is consistent with studies conducted earlier [31–38]. However, we now confirm this in our set of patients. The proportion of unique recombinant forms (URFs) and circulating recombinant forms (CRFs) was substantial at 19% of all isolates. Further investigation of these recombinants is required because of their implications in vaccine design strategies.

Encouragingly, we did not find any transmitted DRM in the 18 cART-naïve patients starting therapy and for whom genotyping was available. Initial transmitted drug resistance (TDR) surveys by the WHO in middle- and low-income countries indicate low-level TDR (<5%) in the majority of surveillance sites and moderate (5–15%) levels of TDR in 17% of sites [39]. Although not following the WHO TDR protocol, our study confirms a low level, as previously reported by others [17,35]. However, two patients had the amino acid 138A in the reverse transcriptase and one patient had 44D. Both mutations are excluded in the WHO list of surveillance DRMs [26] because they also occur as natural polymorphism in drug-naïve patients. None of the treatment-experienced patients harbored isolates with these two mutations. The amino acid 44D has an accessory role in increasing NNRTI resistance if it occurs with thymidine analogue mutations (TAMS) [40]. This mutation occurs in about 1% of isolates from untreated patients but in a significantly higher proportion of patients receiving NRTI [41]. Amino acid 138A has no or little consequence for nevirapine or efavirenz if it occurs on its own [41]. However, it is an important resistance

mutation for rilpivirine [27], a drug not yet available in Tanzania. This mutation has been found in other naïve patients from Tanzania [42]. E138A has been reported as the most prevalent rilpivirine mutation in as many as 3% of drug-naïve patients in the developed world [43]. The mutation was twice as common in a set of viral isolates from various non-B subtypes compared to a set of subtype B isolates. Although not unexpected, we noted a high prevalence of 1.1% in our cohort. This high prevalence of E138A has been shown in other resource-limited settings [44]. However, none of these resistance-related polymorphisms had a clinical impact on first-line therapy in our patients; they all had a GSS of 3.

In total, 22.6% of the patients on their first-line cART for at least six months were failing virologically, and the failure rate was significantly correlated with the duration of therapy. Similar levels have been shown in other resource-limited countries at 24% and 33% of patients treated for a duration of 12 and 24 months, respectively [45]. These levels fail to achieve one of the 90-90-90 targets of achieving 90% viral suppression [46]. At 24–36 months after the start of treatment, there was a high proportion of virological failure compared to other intervals (Fisher's exact test *p*-value = 0.034). However, not enough information is available about potential confounding factors, such as changes in adherence counseling at the study site to perform a rigorous statistical analysis, thus commenting on this difference would be too speculative. In patients with a successful genotype, the majority (82.6%) harbored DRMs. We found several RT DRMs, 69.7% to NRTIs, 82.6% to NNRTIs and 69.7% dual NRTI/NNRTI resistance, consistent with their first-line treatment, which contained zidovudine (AZT) or stavudine (D4T), lamivudine (3TC) and nevirapine (NVP) or efavirenz (EFV), and with other reports in resource-limited settings [45]. In each case, resistance was related to the drugs received. All the protease mutations were so-called "minor DRMs," recognized as natural polymorphisms in the respective subtypes. It has been suggested that polymorphisms in non-B subtypes can affect both the magnitude of resistance conveyed by major mutations as well as the propensity to acquire specific resistance mutations [47,48]. However, our numbers are too small to make any conclusions in this regard.

Among the successful genotypes, the most common mutation was 184V, which was present in most patients on treatment. This mutation confers resistance to lamivudine and emtricitabine. It is also believed to delay the appearance of TAMs [27]. When it occurs together with TAMs, it may cause abacavir (ABC) resistance, one of the second-line drugs in Tanzania at the time of this study. TAMs were also present in a substantial proportion of patients. In one of the patients with more than one follow-up sample, the TAMs occurred later than 184V. The abundance of mutation 103N that confers resistance to efavirenz and nevirapine is of significant concern since these NNRTIs are the mainstay of first-line therapy. Other observed NNRTI mutations were 181C and 190A. All patients failing with resistance had high-level resistance to NNRTIs. Similar mutations were observed in patients from the north part of the country [20]. In our study, the use of stavudine did not lead to mutation K65R. This allows these patients to switch to tenofovir disoproxil fumarate- (TDF) based regimens as a second-line choice. Other studies have indicated the propensity of mutation K65R in subtype C [47,49], but this was not evident from our cohort study.

Concerning protease inhibitor resistance mutations, the polymorphisms 36I, 69K, and 89M were most prevalent, found in one-third or more of all patients. 36I is a common polymorphism in non-B subtypes, while 89M occurs in A, C, F, G, AE, and AG subtypes. The 89M polymorphism can lead to the M89I/L mutation that confers resistance to PIs in various subtypes [47,50,51].

Such a scale of DRMs among failing patients is a critical alert for the country to prepare regimens for the second line. If not controlled, these resistance mutations can spread through transmission, compromising first-line therapy in new infections. The consequence of the resistance is evident in this cohort. Many patients were failing with dual-class resistance. Isolates were resistant to 3TC and other NRTIs, which are the essential components of first-line therapy. That means that these patients need to switch to second-

line, such as ABC or TDF combined with boosted lopinavir (LPV/r) or atazanavir [42,52]. It is noteworthy that the patients in this cohort harbored accessory mutations in the PI region. These mutations are rare in Tanzania, as noted in other studies [18]. These PI mutations may impact the future choice of PI-containing regimens. We tried to predict the failing genotypes' susceptibility to the second-line regimen, where ABC or TDF are combined with FTC and LPV/r. Because of delayed switching, mutations had already accumulated, and GSS to potential second-line therapies was suboptimal (<3) for most failing patients. Considering that fully active boosted PI therapy is scored GSS = 1.5 according to the Rega algorithm used, second-line therapy was already compromised for some patients (GSS ≤ 2). Noteworthily, three patients were already failing with HIVDR on TDF and FTC-based therapy, to which they had been switched for toxicity reasons. These patients had not picked up their pharmacy refills on time. This suggests that patients who are switched to new regimens should be monitored closely for adherence; in this scenario, failure to adhere may jeopardize the future of second-line therapy since they are mostly based on TDF and FTC.

While relatively few patients were failing virologically in our cohort, prevalence and levels of HIVDR in these patients were high hardly nine years after cART scale-up started in Tanzania. We ascribe this to a lack of virological monitoring and adherence counseling. Therefore, apart from the surveillance of HIVDR, it is vital for Tanzania and other RLS to build local capacity to implement viral load and HIVDR testing to guide changes in the standard regimens, reduce the risk of emergence and transmission of HIVDR among patients on treatment, and to implement long-term, successful cART programs effectively. Part of the data gathered in this work will help build such local capacity, develop, test, and improve HIVDR interpretation models. The kind of data gathered here, stored in electronic databases such as the free and open-source RegaDB [24], will allow HIV and AIDS policymakers and healthcare stakeholders to make informed decisions and interventions to mitigate the emergence of drug-resistant HIV isolates among patients.

## 5. Study Limitations

Some of the genotyping was not successful for virologically failing samples. This was probably due to sample degradation. Since the lab performing the assays had no problems with other batches of samples analyzed in the same run, we ascribe this high failure rate to inappropriate storage conditions, even with a viral load of a few hundred copies/ml. Indeed, power failure is a frequent problem, and it is not uncommon for freezers to go through several thawing cycles during the few years the samples were stored until genotyping could be performed. The samples that were successfully genotyped had higher median viral loads—averaging 48,700 (13,980–226,600) copies/ml—than the ones that were not successful—averaging 2449 (824–31,000) copies/ml (*p*-value < 0.01). Viral loads were reassessed for four samples for which genotyping had failed and found undetectable or very low viral loads, suggesting sample deterioration, indeed. As a quality check, baseline and follow-up samples in a few paired sequences were found to cluster together in phylogenetic trees, including appropriate controls [53], confirming that at least these sequences were properly linked per patient.

## 6. Conclusions

These high resistance levels among virologically failing patients call for regular drug-resistance surveillance in Tanzania to control the emergence and transmission of drug resistance in the population.

**Supplementary Materials:** The following are available online at https://www.mdpi.com/article/1 0.3390/microbiolres12040062/s1.

**Author Contributions:** R.Z.S. designed the study, conducted the interviews, performed data and statistical analysis, and wrote the first draft of the manuscript; S.-Y.R. performed data and statistical analysis; P.G., F.M. and R.J.C. supervised laboratory work and manuscript review; E.F.L., E.V.W. and

A.-M.V. supervised the overall study implementation and manuscript development process. All authors have read and agreed to the published version of the manuscript.

**Funding:** R.Z.S. acknowledges the support of the Belgian Technical Cooperation (BTC) for funding his PhD research.

**Institutional Review Board Statement:** We addressed issues of patient confidentiality, benefits, and risks to participating patients, justice, rights and respect that the patients deserve, and the study was approved by the Muhimbili University of Health and Allied Sciences (MUHAS) Research Ethics Committee (MU/DRP/AEC/VOL. XIII/140). Only patients willing to participate in the study and who signed informed consent were recruited into this study. Patient codes were used to delink the patient data in databases. Patients did not receive any payments to motivate them to participate in the study.

**Informed Consent Statement:** Informed consent was obtained from all subjects involved in the study.

**Data Availability Statement:** The dataset analyzed during the current study is available from the corresponding author on request.

**Acknowledgments:** R.Z.S. acknowledges the support of the Belgian Technical Cooperation (BTC) for funding part of this study. A.-M.V. was supported by the Fonds voor Wetenschappelijk Onderzoek Vlaanderen (grant K8.012.12N). Sida funded part of this study under the Muhimbili University of Health and Allied Sciences (MUHAS) small grants scheme by the Fonds voor Wetenschappelijk Onderzoek Vlaanderen (grant G.06.11.09) and by the European Community's Seventh Framework Programme (FP7/2007-2013) under the project "Collaborative HIV and Anti-HIV Drug Resistance Network (CHAIN)" grant agreement no 223131. We are thankful to the patients for cooperating and participating in the study and to the CTC manager, doctors, nurses, counsellors, tracking officers and auxiliary support staff who made the meetings with the patients possible and much more straightforward.

**Conflicts of Interest:** The authors declare no conflict of interest.

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
