# Peer review of "Development of HIV Drug Resistance in a Cohort of Adults on First-Line Antiretroviral Therapy in Tanzania during the Stavudine Era"

_2036-7481, doi:10.3390/microbiolres12040062_

Round 1

Reviewer 1 Report

Major comments:

  1. Since the authors have adherence and baseline VL data in some cases, they should see if these associate with the presence of primary or acquired DR in their cohort. Do the authors have CD4 counts for their cohort? Or other sociodemographic details that can also be considered are potential risk factors for acquired DRM? They should analyze these to present a comprehensive picture.
  2. How many total RT and PR sequences were obtained after sequencing? How many of these were from virologically stable vs. failing patients? Was there any significant difference in DRMs observed from these two groups?
  3. Most of the authors’ cART-naïve patients present with PI mutations, some of which are noted as accessory PI resistance mutations. Is this high mutation rate before therapy initiation been reported from Tanzania before? The authors should discuss these mutations and the rates more in comparison to the rest of the country/continent/subtype, etc.
  4. Ethics clearance and written informed consent statements should also be included in the study cohort section of the methods. Also, the authors note informed consent and not written informed consent. Was this an oversight during manuscript preparation?

Minor comments:

  1. In S Table 1, what is the third time point for patient#27?
  2. Table 2, add median treatment duration for the follow-up data.
  3. How many total RT and PR sequences were obtained after sequencing?

Author Response

Reviewer 1 Comments and response

Dear Reviewer, Dear Editorial Office, below are responses to the comments raised about our manuscript. We have made changes as suggested by reviewers and highlighted yellow in the newly submitted document. We also responded to the comments as shown below.

Point 1: Moderate English changes required.

Response 1: We have edited the document to adjust the grammatical mistakes.

Point 2: Since the authors have adherence and baseline VL data in some cases, they should see if these associate with the presence of primary or acquired DR in their cohort. Do the authors have CD4 counts for their cohort? Or other sociodemographic details that can also be considered are potential risk factors for acquired DRM? They should analyze these to present a comprehensive picture.

Response 2: We thank the reviewer for this kind consideration about using adherence and viral load data to associate with the presence of the mutation. In our two other publications related to the same cohort as the current study, we closely studied the adherence and viral load impact on these patients' treatment outcomes.

The papers are

  1. Sangeda RZ, Mosha F, Prosperi M, Aboud S, Vercauteren J, Camacho RJ, Lyamuya EF, Van Wijngaerden E, Vandamme A-M. Pharmacy refill adherence outperforms self-reported methods in predicting HIV therapy outcome in resource-limited settings. BMC Public Health (2014) 14:1035. doi:10.1186/1471-2458-14-1035
  2. Sangeda RZ, Mosha F, Aboud S, Kamuhabwa A, Chalamilla G, Vercauteren J, Van Wijngaerden E, Lyamuya E, Vandamme A-M. Predictors of non adherence to antiretroviral therapy at an urban HIV care and treatment center in Tanzania. Drug Healthc Patient Saf (2018) Volume 10:79–88. doi:10.2147/DHPS.S143178

While these two studies were powered enough for analytical study because of a large sample size of 162 and 220, respectively. In these two studies, several factors were responsible for poor outcomes. In the first, we noted that of the four types of adherence measurements, the pharmacy refill had the potential to predict virological failure and identify patients to be considered for viral load monitoring and HIVDR testing in resource-limited settings. In the second barrier to adherence included young age and perception of well-being.

However, the smaller final set of patients whose HIV sequences were genotyped in the current study did not attempt an analytical study and conduct associations. Therefore, we have only used a descriptive approach for adherence and viral load in the current manuscript (Table1 and Table 2, respectively). We then referred the readers where necessary to the two papers ([20-21]).

Point 3: How many total RT and PR sequences were obtained after sequencing? How many of these were from virologically stable vs. failing patients? Was there any significant difference in DRMs observed from these two groups?

Response 3: In section 3.3, "The Success Rate of Genotyping," we explain that a total of 41 sequences were available for this study. The criteria for genotyping the HIV isolates was having 1,000 copies per ml in the patient's plasma. We did not analyze virologically stable and failing patients. However, as described in the study limitation, we noted that the samples that were successfully genotyped had higher median viral loads 48,700 (13,980-226,600) copies/ml than the ones that were not successful 2,449 (824 – 31,000) copies/ml (p-value <0.01).

Point 4: Most of the authors' cART-naïve patients present with PI mutations, some of which are noted as accessory PI resistance mutations. Is this high mutation rate before therapy initiation been reported from Tanzania before? The authors should discuss these mutations and the rates more in comparison to the rest of the country/continent/subtype, etc..

Response 4: Thank you for this comment. Indeed we noted the accessory mutations in some patients initiating ART. In the discussion, we have added a paragraph to acknowledge this and that other studies in Tanzania had observed these mutations. In the seventh paragraph in the discussion section, we included the sentences, "It is noteworthy that the patient in this cohort harbored accessory mutations in the PI region. These mutations are rare in Tanzania, as noted in other studies  [17]. These PI mutations may impact the future choice of PIs containing regimens."

 Point 5: Ethics clearance and written informed consent statements should also be included in the study cohort section of the methods. Also, the authors note informed consent and not written informed consent. Was this an oversight during manuscript preparation?

Response 5: We kindly refer to the two sections at the end of the document after conclusions, author contributions and funding section. The sections are not part of the methodology but are labeled Institutional Review Board Statement and Informed Consent Statement. These statements explain how ethical issues and consenting were done.

Point 6: In S Table 1, what is the third time point for patient#27?

Response 6: We have added an explanation below the table that patient number 27 was sampled three times with a note "Samples F176, W0037 and W0116 from patient # 27 the first was taken at study entry and last two after one-year follow-up at one-month interval".

Point 7: Table 2, add median treatment duration for the follow-up data.

Response 7: Patients in this study were followed for one year. Below table 2, the duration of follow-up for therapy experience patients is described as "These patients had been on treatment for a median (IQR) of 32 (22 – 44) months." The methodology explained that a follow-up sample was collected for each patient one year after enrollment.

Point 8: How many total RT and PR sequences were obtained after sequencing?

Response 8: In section 3.3. "The Success Rate of Genotyping" explain that a total of 41 sequences were available for this study. The criteria for genotyping the HIV isolates was having 1,000 copies per ml in the patient's plasma. We did not analyze virologically stable and failing patients. However, as described in the study limitation, we noted that the samples that were successfully genotyped had higher median viral loads 48,700 (13,980-226,600) copies/ml than the ones that were not successful 2,449 (824 – 31,000) copies/ml (p-value <0.01).

Reviewer 2 Report

General comments
Thank you for asking me to review the study entitled ‘Development of HIV drug resistance in a cohort of adults on first-line antiretroviral therapy in Tanzania during the stavudine era’. This study is relevant because HIV drug resistance is one of the major barriers to the success of any HIV treatment program.  Owing to its toxicity profile, global recommendations for the removal of stavudine from national HIV guidelines were established about 10 years ago. Furthermore, a PubMed search using the search term ‘HIV drug resistance AND Tanzania’ yielded a total of 146 articles. Considering this and the progress on anti-retroviral therapy in the last couple of years, the information in this paper provided to the readers appears outdated and may have little impact on the formulation of HIV-related policy. In fact, the authors themselves recognized this fact as they clearly spelled it in the introductory section of the manuscript.

Specific comments

Introduction

This is well written, but there some clarifications are needed.

In lines 70 and 71, the authors stated that ‘The standard second-line regimen consists of before DTG recommendation was lopinavir boosted with ritonavir as the only protease inhibitor (PI) recommended with 2 NRTIs’. This statement emphasized that lopinavir boosted PI was the only second-line drug before the introduction of DTG. This is in contrast to the recommendations of the 2017 Guidelines for management of HIV in Tanzania as highlighted in Chapter 9 (https://www.childrenandaids.org/sites/default/files/2017-04/Tanzania_National-HIV-Guidelines_2015.pdf).

Atazanavir-boosted PI is one of the recommended second-line drugs. The authors should cross-check this information and make the appropriate correction. In lines 79 and 80, the authors cited an article that emphasized that stavudine and zidovudine are still in use in sub-Saharan Africa. I agree this could still apply especially in pediatric HIV. However, the reference cited to back up this information is published in 2017 and it thus appears outdated as there have been several changes in the treatment guidelines since 2017. I could not even find stavudine in Tanzania’s 2017 guidelines. More updated information on this is necessary.

Methods

2019 WHO report on HIV drug resistance stated the two different means by which drug resistance can be assessed.

The pre-treatment drug resistance survey is to provide evidence to inform the selection and effectiveness of first-line treatment and pre-and post-exposure prophylaxis. Assessment of acquired drug resistance (ADR) is performed to provide information regarding the performance of programs and inform the optimal selection of second-and third-line ART. These surveys, therefore, assess viral load suppression and drug resistance in populations receiving ART for 12 (±3) months (referred to as early time point surveys) and ≥48 months (referred to as late time point surveys). The methods highlighted in this paper do not follow any of these two assessment strategies. Participants were randomly selected for enrolment into the study. While it is relevant to provide information to readers on HIV drug resistance, it would be better to provide standardized information in accordance with the WHO guideline on HIV drug resistance for comparison.

In the introductory chapter of this paper, the authors stated the importance of understanding the early warning indicators for the purposes of surveillance of HIV drug resistance. Nonetheless, the authors did not include information on the early warning indicators despite their programmatic relevance.

The authors can also consider adding definitions of terms such as baseline, virological failure, etc.

Results

Lines 208 and 209: ‘During the one-year follow-up, 13 patients had changed therapy because of toxicities to the ARVs’. I suggest that the author use appropriate technical words, such as substitution. In addition, I suggest that the author describe alternatives in the narrative. Authors should consider dividing the results of HIV resistance into pre-treatment and acquired resistance, as this may provide useful information for readers. There is some confusion with what the authors aimed at and what their results are indicating. In the last paragraph of the introductory section (lines 102 to 106), the authors stated that ‘The objective of this study was to document the development of HIVDR during the first-line therapy in Tanzania.  We determined the HIV-1 protease and reverse transcriptase genotypic diversity and drug 103 resistance mutations (DRMs) at study baseline and one-year follow-up, selected from our 104 cohorts, and for whom we had viral load measurements at study baseline or one year of 105 follow-ups’.

The confusion here is that the 2017 Tanzania HIV Guidelines treat PI as a second-line drug. The authors also alluded to this in lines 70 and 71. However, in the results section, the HIV drug resistance mutations for both the first and second lines are documented. The author may consider clarifying this point. In the abstract, the authors stated that 34% of the patients virologically failed antiviral treatment in this study (line 45). Correspondingly, I could not find this information in the results section of the main text. What I could find is that information indicating a detectable viral load in 34% of the patients. The authors may think of clarifying this.

In the abstract, the authors stated that 34% of the patients virologically failed antiviral treatment in this study (line 45). Correspondingly, I could not find this information in the results section of the main text. What I could find is that information indicating a detectable viral load in 34% of the patients. Is this information synonymous with virological failure? Virological failure is clearly defined in Tanzania’s national guidelines and the Consolidated WHO guidelines. I suggest the authors standardize their results' presentation and interpretation in line with national and international guidelines.

Discussion and conclusion

Discussions and conclusions should be consistent with the revised results and methods

Author Response

Reviewer 2 Comments and response

Dear Reviewer, Dear Editorial Office, below are responses to the comments raised about our manuscript. We have made changes as suggested by reviewers and highlighted yellow in the newly submitted document. We also responded to the comments as shown below.

Point 1: English language and style are fine/minor spell check required.

Response 1: We have edited the document to adjust the grammatical mistakes.

Point 2: Thank you for asking me to review the study entitled 'Development of HIV drug resistance in a cohort of adults on first-line antiretroviral therapy in Tanzania during the stavudine era'. This study is relevant because HIV drug resistance is one of the major barriers to the success of any HIV treatment program.  Owing to its toxicity profile, global recommendations for the removal of stavudine from national HIV guidelines were established about 10 years ago. Furthermore, a PubMed search using the search term 'HIV drug resistance AND Tanzania' yielded a total of 146 articles. Considering this and the progress on anti-retroviral therapy in the last couple of years, the information in this paper provided to the readers appears outdated and may have little impact on the formulation of HIV-related policy. In fact, the authors themselves recognized this fact as they clearly spelled it in the introductory section of the manuscript.

Response 2: We thank the reviewer for taking the time to review this manuscript. We agree that stavudine was removed from the treatment guidelines. As we argued in our cover letter associated with this submission, we noted that "Some of the regimens described in our manuscript have been removed from the current treatment guidelines. However, the relevance of patients' profiles on the stavudine backbone stems from the fact that publications of genotyping to determine the HIV drug resistance mutations are rare in Tanzania and other resource-limited settings. The phasing of stavudine in countries happened at various timings such that some countries in Africa have taken longer to withdraw the stavudine-containing regimens completely. Besides, other Nucleotide Reverse Transcriptase Inhibitors' mutations present due to the stavudine-containing regimens such as lamivudine are still part of the newer regimens. Therefore, the mutations may also impact the new regimens. We hope this historical information deserves publication to archive the treatment evolution and the changing regimens in these resource-limited settings. The sequences described in this paper were deposited into GenBank with accession numbers MN816754-MN816797.

We agree that there are publications on HIVDR in PUBMED. However, stricter search criteria such as "HIV Drug Resistance" AND Tanzania return 20 publications, half of which are related to virological failure and review of the topic. Even though the country has more than 1.5 million people living with HIV minority of whom have been put on ART since the late 2000s.

We hope this historical information deserves publication to archive the treatment evolution and the changing regimens in these resource-limited settings.

Point 3: Introduction

This is well written, but there some clarifications are needed.

In lines 70 and 71, the authors stated that 'The standard second-line regimen consists of before DTG recommendation was lopinavir boosted with ritonavir as the only protease inhibitor (PI) recommended with 2 NRTIs'. This statement emphasized that lopinavir boosted PI was the only second-line drug before the introduction of DTG. This is in contrast to the recommendations of the 2017 Guidelines for management of HIV in Tanzania as highlighted in Chapter 9 (https://www.childrenandaids.org/sites/default/files/2017-04/Tanzania_National-HIV-Guidelines_2015.pdf).

Atazanavir-boosted PI is one of the recommended second-line drugs. The authors should cross-check this information and make the appropriate correction. In lines 79 and 80, the authors cited an article that emphasized that stavudine and zidovudine are still in use in sub-Saharan Africa. I agree this could still apply especially in pediatric HIV. However, the reference cited to back up this information is published in 2017 and it thus appears outdated as there have been several changes in the treatment guidelines since 2017. I could not even find stavudine in Tanzania's 2017 guidelines. More updated information on this is necessary.

Response 3: We thank the reviewer, we had corrected the sentence about PI  that "The standard second-line regimen consists of before DTG recommendation was lopinavir or atazanavir boosted with ritonavir as the protease inhibitor (PI) recommended with 2 NRTIs."

Regarding the use of the statement and citation that that stavudine and zidovudine are still in use in sub-Saharan Africa, the 2020 reference (Ndembi, N.; Murtala-Ibrahim, F.; Tola, M.; Jumare, J.; Aliyu, A.; Alabi, P.; Mensah, C.; Abimiku, A.; Quiñones-Mateu, M.E.; Crowell, T.A.; et al. Predictors of first-line antiretroviral therapy failure among adults and adolescents living with HIV/AIDS in a large prevention and treatment program in Nigeria. AIDS Res. Ther. 2020, 17, 64, doi:10.1186/s12981-020-00317-9.) show that 19% were still on stavudine based therapy in Nigeria. We have included this reference.

Point 4: Methods

2019 WHO report on HIV drug resistance stated the two different means by which drug resistance can be assessed.

The pre-treatment drug resistance survey is to provide evidence to inform the selection and effectiveness of first-line treatment and pre-and post-exposure prophylaxis. Assessment of acquired drug resistance (ADR) is performed to provide information regarding the performance of programs and inform the optimal selection of second-and third-line ART. These surveys, therefore, assess viral load suppression and drug resistance in populations receiving ART for 12 (±3) months (referred to as early time point surveys) and ≥48 months (referred to as late time point surveys). The methods highlighted in this paper do not follow any of these two assessment strategies. Participants were randomly selected for enrolment into the study. While it is relevant to provide information to readers on HIV drug resistance, it would be better to provide standardized information in accordance with the WHO guideline on HIV drug resistance for comparison.

In the introductory chapter of this paper, the authors stated the importance of understanding the early warning indicators for the purposes of surveillance of HIV drug resistance. Nonetheless, the authors did not include information on the early warning indicators despite their programmatic relevance.

The authors can also consider adding definitions of terms such as baseline, virological failure, etc.

Response 4: We agree that with the reviewer that the study did not follow the WHO standardized approach to conduct the pre-treatment drug resistance survey and thus making the study rather exploring the proportion of pre-treatment mutations in naïve patients. We agree that future studies need to follow the WHO protocol for surveying pre-treatment drug resistance, including early warning indicators. We have added the definition of virological failure, baseline and follow-up data in section 2.2.3

Point 5: Results Lines 208 and 209: 'During the one-year follow-up, 13 patients had changed therapy because of toxicities to the ARVs'. I suggest that the author use appropriate technical words, such as substitution. In addition, I suggest that the author describe alternatives in the narrative. Authors should consider dividing the results of HIV resistance into pre-treatment and acquired resistance, as this may provide useful information for readers. There is some confusion with what the authors aimed at and what their results are indicating. In the last paragraph of the introductory section (lines 102 to 106), the authors stated that 'The objective of this study was to document the development of HIVDR during the first-line therapy in Tanzania.  We determined the HIV-1 protease and reverse transcriptase genotypic diversity and drug 103 resistance mutations (DRMs) at study baseline and one-year follow-up, selected from our 104 cohorts, and for whom we had viral load measurements at study baseline or one year of 105 follow-ups'.

Response 5: Thanks for these comments. We have rephrased and substituted "had switched therapy" We have divided the results into sections 3.5.1 Pre-treatment Drug Resistance and  3.5.2 Acquired Drug Resistance.

Point 6: The confusion here is that the 2017 Tanzania HIV Guidelines treat PI as a second-line drug. The authors also alluded to this in lines 70 and 71. However, in the results section, the HIV drug resistance mutations for both the first and second lines are documented. The author may consider clarifying this point. In the abstract, the authors stated that 34% of the patients virologically failed antiviral treatment in this study (line 45). Correspondingly, I could not find this information in the results section of the main text. What I could find is that information indicating a detectable viral load in 34% of the patients. The authors may think of clarifying this.

Response 6: Thank you for this comment. The 2017 guideline recommends ATV and LPV boosted with ritonavir. However, in practice, LPV/r is the most supplied PI in Tanzania's care and treatment centers. When showing the GSS score, we try to predict whether, given the current mutations, which first or the second line will render the HIV isolate susceptible during treatment.

Concerning the virologically failed antiviral treatment, virological failure was interchangeably used with a detectable viral load where the detection limit was 400 copies/ml in the old Tanzania HIV treatment guidelines.

Point 7: In the abstract, the authors stated that 34% of the patients virologically failed antiviral treatment in this study (line 45). Correspondingly, I could not find this information in the results section of the main text. What I could find is that information indicating a detectable viral load in 34% of the patients. Is this information synonymous with virological failure? Virological failure is clearly defined in Tanzania's national guidelines and the Consolidated WHO guidelines. I suggest the authors standardize their results' presentation and interpretation in line with national and international guidelines.

Response 7: About the virologically failed antiviral treatment, it was interchangeably used with detectable viral load where the detection limit was 400 copies/ml in the old Tanzania HIV treatment guidelines.

Point 8: Discussions and conclusions should be consistent with the revised results and methods

Point 8: We thank the reviewer, we have considered the suggested changes and reflected them in the discussion and conclusion.

Reviewer 3 Report

The manuscript describes a study of genotypic resistance, detected in a rural setting in Tanzania before and after a limited period of first line therapy.

The study is quite limited in size, and a high proportion of included samples were unsuited for analysis. However, since this is a typical situation for such African setting, even this described aspect is useful for related studies. Nevertheless, the small number of resistances and the small basis for the frequency calculation should be clearly emphasised in the discussion of those points (i.e. line 370f)

Points requiring attention:

main:

  • why does Figure 2 show a significantly(?) higher failure rate for months 24-36? This is not evident and not discussed by the authors. Please list possible reasons.
  • line 370: the NNRTI mutation 138A is indicated as "worrisome". Please discuss the findings of Sluis-Cremer et al, Antivir.Res 2014, 107:31, who show that >6% of subtype C and >8% of NNRTI-naive samples may carry this mutation. The statement should be corrected and rephrased in the text!
  • line 413: by the way, 3TC is called an NRTI - correct phrasing.
  • lines 423-425: the sentence erroneously implies that adherence is mainly an issue of TDF/FTC therapy. Change phrasing to avoid this impression!
  • lines 427-431: multiple studies have demonstrated(!) that the main issue is NOT primarily the "virological monitoring" that is needed but rather adherence! Please consider recent studies from rural Lesotho as reference! Please rephrase and correct the statement

minor:

line 98: is it true that in this setting in Dar, patients have to return MONTHLY for an ARV pickup? It is standard in other Tanzanian settings that the spacing is rather 3or 6 months, depending on stability of therapy!

line 165: sentence is incomplete/truncated - please correct!

line 311: (table 3) the line of 89I under PI has a value of 4.35% - with no case?? - please correct or explain

line 317 (Table 4) the reader may prefer to see 3TC and FTC next to each other, since resistances to both drugs have the identical patterns

--> grammar and language will need some attention of an experienced native speaker, e.g.

line 100 - genotypic resistance testing instead of "genotypic studies)

line 350 - remove the 'comma'

Author Response

Reviewer 3 Comments and response

Dear Reviewer, Dear Editorial Office, below are responses to the comments raised about our manuscript. We have made changes as suggested by reviewers and highlighted yellow in the newly submitted document. We also responded to the comments as shown below.

Point 1: Moderate English changes required.

Response 1: We have edited the document to adjust the grammatical mistakes.

Point 2: The manuscript describes a study of genotypic resistance, detected in a rural setting in Tanzania before and after a limited period of first line therapy.

The study is quite limited in size, and a high proportion of included samples were unsuited for analysis. However, since this is a typical situation for such African setting, even this described aspect is useful for related studies. Nevertheless, the small number of resistances and the small basis for the frequency calculation should be clearly emphasised in the discussion of those points (i.e. line 370f)

Response 2: We thank the reviewer for the comment. We have changed the sentence at line 370 to reflect the numbers.

Point 3: why does Figure 2 show a significantly(?) higher failure rate for months 24-36? This is not evident and not discussed by the authors. Please list possible reasons.

Response 3:

The difference in virological failure rates at different ART duration was significant (Fisher's Exact Test p-value = 0.034).

However, we do not have enough information to judge the confounding factors, such as changes in adherence counseling at the study site. Therefore we refrain from speculation on the possible explanations for a high proportion of virological failure at 24-36 months after the start of treatment compared to other intervals.

We have added this statement to the manuscript discussion "At 24-36 months after the start of treatment, there was a high proportion of virological failure compared to other intervals (Fisher's Exact Test p-value = 0.034), however not enough information is available about potential confounding factors such as changes in adherence counseling at the study site to perform a rigorous statistical analysis, and commenting on this difference would be too speculative".

Point 4: line 370: the NNRTI mutation 138A is indicated as "worrisome". Please discuss the findings of Sluis-Cremer et al, Antivir.Res 2014, 107:31, who show that >6% of subtype C and >8% of NNRTI-naive samples may carry this mutation. The statement should be corrected and rephrased in the text!

Response 4: We have changed the reference and cite the reference Sluis-Cremer N, Jordan MR, Huber K, Wallis CL, Bertagnolio S, Mellors JW, Parkin NT, Richard Harrigan P. E138A in HIV-1 reverse transcriptase is more common in subtype C than B: Implications for rilpivirine use in resource-limited settings. Antiviral Res (2014) 107:31–34. doi:10.1016/j.antiviral.2014.04.001.

Point 5: line 413: by the way, 3TC is called an NRTI - correct phrasing.

Response 5: We have rephrased to "3TC and other NRTIs."

Point 6: lines 423-425: the sentence erroneously implies that adherence is mainly an issue of TDF/FTC therapy. Change phrasing to avoid this impression!

Response 6: We have modified the statement slightly to mean that patients switching to new therapies need to be closely monitored so that they do not develop resistance to the backbone of second line treatment.

Point 7: lines 427-431: multiple studies have demonstrated(!) that the main issue is NOT primarily the "virological monitoring" that is needed but rather adherence! Please consider recent studies from rural Lesotho as reference! Please rephrase and correct the statement

Response 7: We have added that the development of resistance may be due "to a lack of virological monitoring and adherence counseling."

Point 8: line 98: is it true that in this setting in Dar, patients have to return MONTHLY for an ARV pickup? It is standard in other Tanzanian settings that the spacing is rather 3or 6 months, depending on stability of therapy!

Response 8: We have rephrased to reflect that patients return centers three to six monthly intervals for a refill.

Point 9: line 311: (table 3) the line of 89I under PI has a value of 4.35% - with no case?? - please correct or explain

Response 9: case (1) was added and the proportion corrected.

Pont 10: line 317 (Table 4) the reader may prefer to see 3TC and FTC next to each other, since resistances to both drugs have the identical patterns

Response 10: Sorry, this comment was not very clear. The line numbers may have changed while saving the pdf file.

Point 11: --> grammar and language will need some attention of an experienced native speaker, e.g. line 100 - genotypic resistance testing instead of "genotypic studies). line 350 - remove the 'comma'

Response 11: These sentences were changed.

Reviewer 4 Report

The aim of this study is to examine the development of drug resistance in a cohort in Tanzania. This work was performed and evaluated by genotyping the virus from samples from patients at baseline and in follow-up samples. Examination of the presence of virus after ART indicated that there was a portion of participants who were unresponsive to treatment. Drug resistance was further examined by the evaluation and prevalence of drug resistance mutations. 

The strengths of the paper is the evaluation of the subtype diversity of the cohort and the HIV drug resistance mutation profiles of the cohort. The discussion was also well written and the discussion regarding treatment and its impact on the drug resistance profiles observed is important. I agree with the authors that a database regarding treatment options, and outcomes regarding plasma viral load and drug resistance is critical in mitigating the emergence of HIV drug-resistant isolates. 

The main weakness of the paper is that this information may not be novel to the HIV field as a whole. Although, I do agree that this information may be important for the areas where there is a limitation to access of all available HIV treatment options. 

Line 414 - Patents should be patients. 

Author Response

Reviewer 4 Comments and response

Dear Reviewer, Dear Editorial Office, below are responses to the comments raised about our manuscript. We have made changes as suggested by reviewers and highlighted yellow in the newly submitted document. We also responded to the comments as shown below.

Point 1: English language and style are fine/minor spell check required.

Response 1: We have edited the document to adjust the grammatical mistakes.

Point 2: The aim of this study is to examine the development of drug resistance in a cohort in Tanzania. This work was performed and evaluated by genotyping the virus from samples from patients at baseline and in follow-up samples. Examination of the presence of virus after ART indicated that there was a portion of participants who were unresponsive to treatment. Drug resistance was further examined by the evaluation and prevalence of drug resistance mutations. 

The strengths of the paper is the evaluation of the subtype diversity of the cohort and the HIV drug resistance mutation profiles of the cohort. The discussion was also well written and the discussion regarding treatment and its impact on the drug resistance profiles observed is important. I agree with the authors that a database regarding treatment options, and outcomes regarding plasma viral load and drug resistance is critical in mitigating the emergence of HIV drug-resistant isolates. 

The main weakness of the paper is that this information may not be novel to the HIV field as a whole. Although, I do agree that this information may be important for the areas where there is a limitation to access of all available HIV treatment options. 

Line 414 - Patents should be patients. 

Response 2: We thank the reviewer for his positive comments. The words patents have been replaced with patients

As we argued in our cover letter associated with this submission, we noted that "Some of the regimens described in our manuscript have been removed from the current treatment guidelines. However, the relevance of patients' profiles on the stavudine backbone stems from the fact that publications of genotyping to determine the HIV drug resistance mutations are rare in Tanzania and other resource-limited settings. The phasing of stavudine in countries happened at various timings such that some countries in Africa have taken longer to withdraw the stavudine-containing regimens completely. Besides, other Nucleotide Reverse Transcriptase Inhibitors' mutations present due to the stavudine-containing regimens such as lamivudine are still part of the newer regimens. Therefore, the mutations may also impact the new regimens. We hope this historical information deserves publication to archive the treatment evolution and the changing regimens in these resource-limited settings. The sequences described in this paper were deposited into GenBank with accession numbers MN816754-MN816797.

Round 2

Reviewer 1 Report

The authors have provided explanations and details as requested and the revised version is satisfactory.

Reviewer 2 Report

No additional comments